# Identification of Novel Genes Associated with Fish Skeletal Muscle Adaptation during Fasting and Refeeding Based on a Meta-Analysis

**DOI:** 10.3390/genes13122378

**Published:** 2022-12-16

**Authors:** Érika Stefani Perez, Sarah Santiloni Cury, Bruna Tereza Thomazini Zanella, Robson Francisco Carvalho, Bruno Oliveira Silva Duran, Maeli Dal-Pai-Silva

**Affiliations:** 1Department of Structural and Functional Biology, São Paulo State University (UNESP), Botucatu 18618-689, Brazil; 2Department of Histology, Embryology and Cell Biology, Federal University of Goias (UFG), Goiania 74690-900, Brazil

**Keywords:** skeletal muscle growth, fish, pacu, *Piaractus mesopotamicus*, fasting, refeeding, meta-analysis

## Abstract

The regulation of the fish phenotype and muscle growth is influenced by fasting and refeeding periods, which occur in nature and are commonly applied in fish farming. However, the regulators associated with the muscle responses to these manipulations of food availability have not been fully characterized. We aimed to identify novel genes associated with fish skeletal muscle adaptation during fasting and refeeding based on a meta-analysis. Genes related to translational and proliferative machinery were investigated in pacus (*Piaractus mesopotamicus*) subjected to fasting (four and fifteen days) and refeeding (six hours, three and fifteen days). Our results showed that different fasting and refeeding periods modulate the expression of the genes *mtor*, *rps27a*, *eef1a2*, and *cdkn1a*. These alterations can indicate the possible protection of the muscle phenotype, in addition to adaptive responses that prioritize energy and substrate savings over cell division, a process regulated by *ccnd1*. Our study reveals the potential of meta-analysis for the identification of muscle growth regulators and provides new information on muscle responses to fasting and refeeding in fish that are of economic importance to aquaculture.

## 1. Introduction

### 1.1. Fish and Skeletal Muscle

Fish are the most diverse living vertebrates on the planet, with important ecological and economic roles in many nations [1]. Regarding the fishing industry in Brazil, *Piaractus mesopotamicus*, popularly known as the pacu, is one of the most widely cultivated native species, with production in nine Brazilian states [2]. The popularity of pacu is associated with certain characteristics of the species, such as its robustness, high carcass yield, and good adaptation to artificial feeding [3]. In addition, improvements in management techniques developed through studies seeking to understand how aspects of the cultivation environment affect the development of this species have contributed to the high level of pacu production [4]. Recently published in vivo and in vitro studies explored the genetic attributes that regulate the fast growth and impressive musculature of this species [5,6], with the pacu presenting an excellent model for studies of the skeletal muscle [7]. 

The skeletal muscle constitutes more than 60% of the teleost body mass [8]. In addition to being one of the most frequently consumed tissues by humans, the skeletal muscle acts as a support system for the body, generating strength and movements, and represents a large reserve of amino acids for fish [9]. When faced with environmental stimuli, the muscle proteins (myofibrillar components) are degraded or synthesized, enabling muscle plasticity. The predominance of anabolic pathways in the tissue promotes fish muscle growth by increasing the size of the existing muscle fibers (hypertrophy) or forming new muscle fibers (hyperplasia). However, muscle loss occurs when catabolism exceeds protein synthesis [10,11]. In this context, the availability of nutrients is a factor that stimulates the remodeling of the skeletal muscle, given that the availability of food in nature changes over time and space [12]. Fasting periods are common in nature and fish farming, where periods of food restriction followed by refeeding are used as a compensatory growth strategy [13]. 

Fasting and refeeding studies conducted in our lab showed changes in the classical pathways that regulate the muscle protein balance in the pacu skeletal muscle, which is well-characterized in the literature [14,15,16]. First, during fasting, there is an increase in the expression of catabolic genes (*murf1* and *mafbx*) and a loss of muscle mass. Then, during refeeding periods, muscle mass recovery occurs through the stimulation of anabolic genes (*igf-1* and *mtor*) [17]. These observations are similar to those of studies carried out on *Hippoglossus hippoglossus* [18], *Salmo salar* [19], *Oncorhynchus mykiss* [20], and *Paralichthys adspersus* [21], demonstrating the effectiveness of these experimental protocols in observing the mechanisms of changes in the muscle phenotype. However, the regulators of the muscle phenotype in fish have not yet been fully identified, especially under challenging conditions. 

### 1.2. Meta-Analysis

In finfish aquaculture, the maintenance of the fish muscle mass underpins the sector’s profitability. Thus, understanding the muscle phenotype regulators is essential in order to improve the management techniques. In this context, mRNA profiles in public databases may favor the exploration of markers involved in maintaining the skeletal muscle phenotype and its growth. Among the available methodologies for the mining of the public data to be analyzed, meta-analysis has been used in several areas of science as a statistical method that compiles, compares, and evaluates data from pre-existing and independent studies [22,23]. This methodology has already been used to study fish skeletal muscle in analyses of the metabolic capacity associated with the reduction in nutrients [24], to evaluate the seasonal variations in mercury concentration [25], and to recognize pacu as a species of fish with a high muscular expression of myoglobin [26]. However, to our knowledge, no meta-analysis studies have yet focused on evaluating the possible muscle growth mediators or the mechanism maintaining the muscle phenotype in fish. 

Therefore, this work aimed to identify, through meta-analysis, novel genes associated with the growth and maintenance of the muscle phenotype. We investigated the expression levels of these genes during fasting and refeeding conditions in the pacu fish, *Piaractus mesopotamicus*. 

## 2. Materials and Methods

### 2.1. In Silico Analysis

#### 2.1.1. Integrative Meta-Analysis

The integrative meta-analysis was performed according to the Preferred Reporting Items for Systematic Reviews and Meta-Analyses (PRISMA, http://www.prisma-statement.org/, accessed on 2 February 2020) [27]. The keywords used to search for the studies were: “fish AND skeletal muscle”, “fish skeletal muscle AND transcriptome”, “fish muscle AND transcriptomic”, “fish skeletal muscle AND transcriptomics”, and “fish skeletal muscle AND mRNA profiling”. The meta-analysis involved studies carried out between 2012 and 2016. The inclusion criteria were (1) studies using raw data on mRNAs from the white and red muscles of fish and (2) samples derived from fish of the Osteichthyes class alone. The exclusion criteria were (1) studies using genetic manipulation, (2) studies that did not use fish as a model, (3) studies using models of chronic disease, (4) studies that did not assess the skeletal muscle, (5) single-cell data, (6) cell culture data, (7) studies that did not contain transcriptomic data, (8) aging models, (9) studies based on fish of the Chondrichthyes class, and (10) studies based on insufficient differentially expressed genes (DEGs) or less than ten genes (Appendix A). The GEO database’s datasets (NCBI) were used to search for and download skeletal muscle mRNA data from the included studies (Figure 1).

#### 2.1.2. Analysis of the Microarray Data and Differentially Expressed Genes (DEGs)

Studies involving microarray data and their respective serial numbers were obtained from the GEO datasets. Since microarray experiments can be designed on two different platforms, including one-color or two-color arrays, specific analyses were applied for data from each platform. The raw data of the GSE84288 and GSE58929 datasets and the DEGs of GSE36339 and GSE47141 were downloaded and analyzed. For the analysis of these data in the one-color system, the GEO2R tool (http://www.ncbi.nlm.nih.gov/geo/geo2r/, accessed on 1 June 2020) was used, which applies the same script (R 3.2.3, Biobase 2.30.0, GEOquery 2.40.0, limma 3.26.8) to all the datasets. To select the differentially expressed genes (DEGs) in this system, adjusted *p*-values (padj) of <0.05 and a |fold change (FC)| of >1.5 were considered (Appendix A). For the analysis of the data in the two-color system [28], the DEGs were obtained through a list provided by the authors. They applied the difference in the log2 ER (expression ratio: Cy5/Cy3 or Cy3/Cy5) of zero between the chips to obtain these DEGs from this list, and we considered *p* < 0.05 for our DEGs (Appendix A). Cy5 and Cy3 are cyanines that emit fluorescence when there is a hybridization between the cDNA of the sample and the probe contained in the microarray chip. 

#### 2.1.3. Ontologies and Network of Molecular Interactions

Based on the DEGs of each dataset, the enrichment of the pathways was performed using the Fish EnrichR tool (https://amp.pharm.mssm.edu/FishEnrichr/, accessed on 15 July 2020). In this analysis, the lists of up- and downregulated genes were combined using the tool due to the similarity of the biological processes in which these genes take part. The Biological Process, Molecular Function, WikiPathway, and KEGG libraries were considered for the selection of the terms, considering the significance value of *p* < 0.05 (Fisher’s test) (Appendix A). Based on the ontology of each dataset, an initial and individual analysis was performed to identify the most common pathways among the datasets (Appendix A). These common pathways, being present in at least two datasets (Appendix A), had their genes selected for the construction of a molecular interaction network using the STRING version 11 tool (https://string-db.org/, accessed on 25 August 2020). The databases, text mining, experiments, co-expression, and co-occurrence were considered as active interaction sources. The minimum interaction score required was the most reliable (0.900), and the *p*-value of the PPI enrichment indicated the statistical significance provided by STRING (<1.0 × 10^−16^). For a better visualization, the nodes without connections were hidden, and the names of the proteins were retained. Based on this network and with the support of the literature, key genes were selected for the investigation of the muscle of pacus (*Piaractus mesopotamicus*) subjected to fasting and refeeding conditions through RT-qPCR. 

### 2.2. In Vivo Analysis

#### 2.2.1. Ethics Statement and Animals

The experiments were carried out on animals and their muscle tissue from previous studies [6,16] according to the ARRIVE [29] guidelines and those of the Brazilian National Council for the Control of Animal Experimentation (CONCEA). The protocol was approved by the Ethics Committee on Animal Use (protocol numbers 705 and 1050) of the Institute of Biosciences, São Paulo State University (UNESP), Botucatu, São Paulo, Brazil. Early juvenile pacus weighing 12.6 g ± 2.24 and late juvenile pacus weighing 28.5 g ± 8.8 were obtained from the University of West São Paulo (UNOESTE), Presidente Prudente, São Paulo, Brazil, and cultivated at the Agribusiness Technology Agency (APTA), Presidente Prudente, São Paulo, Brazil. The fish were reared at 28 °C under a natural photoperiod (12 h light:12 h dark) in different recirculation systems in 0.5 m^3^ tanks containing 30 pacus. All the fish underwent a 15-day adaptation period with ad libitum food two to three times a day, following the diet previously described in the publications of our research group [6,16], before the start of the experiments. 

After this adaptation period, the youngest fish (12.6 g ± 2.24) were separated into the following groups: control (C, with ad libitum feeding), fasting (F4d, fasting for four days) and refeeding (R3d, refeeding for three days after fasting). Meanwhile, the older fish (28.5 g ± 8.8) were divided into the following groups: control (C, with ad libitum feeding), fasting (F15d, fasting for 15 days), and refeeding (R6h, refeeding for six hours after fasting; R15d, refeeding for 15 days after fasting). Fish of different weights were selected to ensure that the investigations would cover different growth periods. In this context, the lengths of the fasting and refeeding periods were selected based on the size of the fish, considering that these nutritional changes would modify the muscle transcriptional signature without triggering the death of the fish before their experimental evaluation. In addition to these aspects, our research group has already obtained significant results by evaluating the classical genes of the skeletal muscle in these periods of fasting and refeeding in pacus [6,16], suggesting that evaluations of novel genes in these experimental models can be of great value for the area. After the experiments, the pacus were euthanized with excess benzocaine (Sigma Aldrich, St. Louis, MO, USA) at a concentration up to and exceeding 250 mg/L, and the fast-twitch muscles were collected from each experimental group (5 fish/experimental group). Afterward, these muscles were stored in a freezer at −80 °C until the beginning of the molecular analysis. 

#### 2.2.2. mRNA Expression

To investigate the genes selected in the in silico analysis, we extracted the total RNA from pacu muscle samples previously used in other experiments [6,16] using TRIzol^®^ reagent (Invitrogen Life Technologies, Waltham, MA, USA), following the manufacturer’s instructions. To quantify and analyze the purity of the extracted RNA, a NanoVue^TM^ Plus spectrophotometer (GE Healthcare, Chalfont St Giles, UK) was used, which provides these parameters through absorbance at 260 nm (amount of RNA) and 280 nm (amount of protein). To assess the integrity of the extracted RNA, 1% agarose gel electrophoresis and capillary electrophoresis were used with the 2100 Bioanalyzer (Agilent, Santa Clara, CA, USA) were used, which provided an RNA integrity number (RIN) based on 28S and 18S ribosomal RNAs. The samples with a 260/280 ratio of ≥1.8 and RIN of ≥7 were considered for further analysis. To prevent any genomic DNA from contaminating the samples, the extracted RNA was treated with RQ1 RNase-Free DNase (Promega, Madison, WI, USA). RNA reverse transcription was performed using a GoScript^TM^ Reverse Transcription System (Promega, USA), according to the manufacturer’s guidelines. 

The expression level of the mRNAs was evaluated by quantitative real-time PCR (qPCR) using the QuantStudio™ 12K Flex Real-Time PCR platform (Thermo Fisher Scientific, USA), with two technical and five biological replicates. All the qPCR reactions were performed according to the Minimum Information Guidelines for Publishing Quantitative Real-Time PCR Experiments (MIQE) [30]. The cDNA samples were amplified using a GoTaq^®^ qPCR Master Mix (Promega, USA), and the primers for the respective targets (Table 1) were synthesized using Exxtend (BRA). The primers for *rps27a* (ribosomal protein s27a), *eef1a2* (eukaryotic translation elongation factor 1 alpha 2), *ccnd1* (Cyclin D1), *mtor* (the mammalian target of rapamycin), and *cdkn1a* (cyclin-dependent kinase inhibitor 1A) were designed based on the *Piaractus mesopotamicus* transcriptome obtained by our research group [5]. The tools used to design the primers were Primer3 v.0.4.067,68 (https://bioinfo.ut.ee/primer3-0.4.0/, accessed on 20 October 2021) and NetPrimer (Premier Biosoft, San Francisco, CA, USA) (Table 1). The reactions were performed at 95 °C for 10 min, followed by 40 cycles of denaturation at 95 °C for 15 s and annealing at 60 °C for 1 min. For the specific genes, such as *eef1a2* and *cdkn1a*, the annealing temperature was changed to 65 °C so as to improve the reaction. The relative quantification of the gene expression was performed by the comparative CT method, 2^ΔΔCt^ [31] using DataAssist^TM^ v3.01 software (Thermo Fisher Scientific, Waltham, MA, USA), in which the expression was normalized based on the *ppia* mRNA, the expression of which remained constant among all the samples.

### 2.3. Statistical Analyses

For the gene expression analysis, comparisons were performed between the experimental groups of late juveniles C, F15d, R6h, and R15d, respectively, and between the early juvenile groups C, F4d, and R3d. The Kolmogorov–Smirnov normality test was performed for all the genes evaluated, and one-way ANOVA tests with Tukey’s post-test were applied to the data with a normal distribution. The Kruskal–Wallis test and Dunn post-test were applied to the data that did not have a normal distribution. The data are presented as the mean ± SD, and for all the tests, the statistical significance was set at 5% (*p* < 0.05) (software: GraphPad v.6 (GraphPad Software, La Jolla, CA, USA, www.graphpad.com, accessed on 26 November 2021).

## 3. Results

### 3.1. In Silico Analyzes

Four datasets using microarrays to study the gene expression were selected based on our meta-analysis. These datasets have their respective serial numbers identified in the database as GSE58929, GSE84288, GSE36339, and GSE47141. Fish of the *Danio rerio* species subjected to exercise, *Salmo salar* that consumed glucosinolates, *Sparus aurata* that consumed lipopolysaccharides, and *Oncorhynchus mykiss* subjected to exercise and a high-carbohydrate diet were the models of these studies, respectively. In total, 1906 upregulated genes and 617 downregulated genes were identified. For the GSE58929 dataset, 1516 upregulated genes and 215 downregulated genes were obtained, with 195 upregulated and 158 downregulated genes for GSE84288, 43 upregulated and 38 downregulated genes for GSE36339, and, finally, 152 upregulated and 206 downregulated genes for GSE47141 (Table 2). For the enrichment analysis, lists containing both the up- and downregulated genes of each dataset were used. The cellular processes most enriched in each dataset were mainly associated with ribosomes and the cell cycle and represented by terms such as cytoplasmic translation (GO:0002181), cytoplasmic ribosomal proteins_WP324, ribosome, positive regulation of the cell cycle (GO:0045787), and cell cycle_WP1393 (Appendix A). These enriched terms, which are associated with protein synthesis and proliferation and were present in at least two datasets, were identified (Figure 2). Thus, the genes were selected and plotted in a molecular interaction network (Figure 3). Consequently, the genes obtained from the GSE84288 dataset were not present in the composition of the network, as they enriched cellular processes that did not match the objectives of the study. 

In the interaction network, we observed different clusters of genes associated with the cell cycle (upper region of the network) and ribosomes (lower region), demonstrating the interaction between these two processes. Furthermore, such genes were responsible for enriching the common cellular processes of proliferation and protein synthesis in the different species subjected to different treatments (Figure 2). 

From this network, key genes were selected by considering their topology and the possible importance of these genes for fish skeletal muscles, according to the literature. This selection involved *rps27a*, which connects the two large clusters of the network, and based on their close interactions, the other targets selected were *ccnd1*, *cdkn1a*, and *eef1a2*. The selected key genes were investigated using the fasting and refeeding conditions, which involve proliferation and protein synthesis processes and have great relevance in aquaculture.

### 3.2. mRNA Expression

#### 3.2.1. Early Juvenile Pacus Subjected to Four Days of Fasting (F4d) and Three Days of Refeeding (R3d)

Our evaluation of the shorter fasting and refeeding periods showed an increase in the *mtor* and *eef1a2* expression in the skeletal muscle of the early juveniles in the F4d group compared to the C and R3d groups, while the expression of *rps27a* was reduced in the R3d group compared to the F4d group. Furthermore, considering the genes associated with the cell cycle, a reduction in the *ccnd1* expression was observed in the R3d group compared to both the C and F4d groups. The expression of *cdkn1a*, meanwhile, was downregulated in the R3d group compared to F4d (Figure 4).

#### 3.2.2. Late Juvenile Pacus Subjected to Fasting for 15 Days (F15d) and Then Refeeding for Six Hours (R6h) or 15 Days (R15d)

Considering the genes related to the translational machinery, at the end of the 15-day fasting period (F15d), the *mtor* expression remained unchanged in the skeletal muscle of the late juvenile pacus compared to the C group. On the contrary, the *mtor* expression was reduced in the prolonged refeeding group (R15d) compared to the R6h group. Regarding the *rps27a* ribosomal gene, there was a decrease in its expression in the R15d group compared to the F15d group. On the other hand, the expression of the eukaryotic elongation factor *eef1a2* showed no change in its expression between the different groups. Regarding the cell cycle and replication regulators, the expression of the *ccnd1* transcripts showed an increase in the F15d group and the R6h group compared to the C and R15d groups. Meanwhile, the expression of *cdkn1a* was downregulated in both the R6h and R15d groups compared to the C and F15d groups (Figure 5).

## 4. Discussion

The meta-analysis using the fish skeletal muscle microarrays allowed us to identify the critical connection between ribosomes and the cell cycle, as evidenced by the molecular interaction network. This relationship was also observed by a group of researchers who described the limiting role of the ribosomal machinery in cell proliferation. This association becomes more evident when considering the roles of many ribosomal proteins (RPs) in communicating the disturbance in ribosome biogenesis to the cell cycle [32]. Furthermore, the lack of nutrients and other factors promotes ribosomal/nucleolar stress, characterized by the destabilization of the nucleolus and interruption of the synthesis of new ribosomes, triggering changes in the cell physiology, such as the arrest of the division process [33]. In such a context, we investigated the effects of fasting and refeeding on the expression of the regulators and components of the translational and proliferative machinery obtained in the in silico analyses with implications for muscle growth in fish. 

In addition to the other targets obtained and investigated in this work that are related to ribosomes, such as *rps27a* and *eef1a2*, mTOR (the mechanistic target of rapamycin kinase) plays an essential role in ribosomal biogenesis by regulating the translation of RPs and the transcription of rRNAs (ribosomal RNAs) [34]. Although this gene was not identified in the interaction network, based on its importance for the cellular process studied and to better understand the findings obtained in silico, the expression of the *mtor* transcript was also evaluated. Based on this analysis, the upregulation of *mtor* in the F4d compared to the C group may be involved in maintaining the translational capacity, defined as the number of ribosomes present in the tissue [35], during this initial period of fasting. This maintenance can be seen as a mechanism that supports the ribosomal population and basal protein synthesis and protects the muscle tissue during the initial stress triggered by the four-day fasting [15,36,37]. Thus, this transcriptional modulation may occur until the time of food restitution, as observed by the expression of *mtor* in the R3d group, the levels of which became similar to those of the control group. On the other hand, the lack of alteration in the expression of this gene in the F15d group compared to the control may be related to the more severe catabolic environment compared to the F4d condition. In this group, it is possible that the processes maintaining the survival of the organism, as a whole, overcome the protection of the muscle mass.

Interestingly, the expression of *rps27a* was similar when comparing F4d with R3d and F15d with R15d. In all cases, the expression of this gene increased during fasting. Similar to other RPs, RPS27A plays an important structural role in the ribosomes, and its participation in the crosstalk between ribosomal stress and the cell cycle has already been described [32]. According to these authors, who evaluated the expression of RPL11 (ribosomal protein L11) in vitro, its expression was unchanged despite the role that this RP plays in nutritional stress [38]. In addition to these findings, the authors noted that it was abundant even in cells that were not subjected to starvation [39]. Our data indicated that *rps27a* is also abundant in the muscle cells, demonstrating the role of the translation machinery in the tissue and the presence of multiple copies of rDNA in the skeletal muscle [40]. Furthermore, a reduction in *rps27a* expression was observed in the R3d and R15d groups compared to F4d and F15d, respectively. These results may be associated with the reduction in stress in the context of food availability in the refeeding groups, as was also observed in the refeeding of *Gadus morhua* after fasting periods, which promoted the resumption of ribosomal pro-biogenesis conditions in the muscle of this fish [41]. 

The expression of *eef1a2* (eukaryotic translation elongation factor 1 alpha 2), another component of the translational machinery transcript, was also evaluated in our study. In its canonical function, *eef1a2* acts through the elongation of the polypeptide chains, ensuring that the tRNAs and their respective amino acids correctly couple on the ribosomes [42]. In this study, four-day fasting promoted an increase in *eef1a2* expression, followed by a reduction in the expression of the R3d group. According to some authors, Eef1a2 participates in the maintenance of muscle mass in rats, and its absence triggers atrophy [43,44]. Similarly, in zebrafish embryos, the lack of this gene affects the development of nervous tissue and causes muscle weakness [45].

Furthermore, in *mdx* mice, eef1a2 overexpression improves Duchenne muscular dystrophy, highlighting the relevance of this gene to tissue maintenance, formation, and morphology [46]. Given the critical role of *eef1a2* in the conservation of the muscle mass and morphology, the increase in its expression in the F4d group may indicate an attempt to maximize protein synthesis during food restriction. In contrast, the food availability in the R3d group reduced its expression to levels similar to those of the C group. These data, as well as the lack of alteration in *eef1a2* in the more extended periods of fasting and refeeding, corroborate the transcriptional modulation of *mtor*. In the shorter periods of fasting and refeeding, it was still viable for cells to maintain the muscle mass through the utilization of amino acids from muscle catabolism during the short protocol [15,47], something that was unlikely to occur in the F15d group. 

As in the case of the genes associated with the protein synthesis machinery, changes in the nutritional status of cells also modulate the expression of cell cycle regulators [48]. *Ccnd1* (cyclin D1) is one of the genes that coordinate cell proliferation, connecting nutrient availability in the environment with cell division. The expression of this cyclin varies throughout the cell cycle, with high levels of *ccnd1* found in the G1 phase. In the transition from the G1 to S phases, there is a reduction in cyclin expression, which allows for DNA synthesis. During the entire S phase, the *ccnd1* levels remain low, thus increasing the cyclin expression in the transition from S to G2. At this point, the cyclin D1 levels only increase due to environmental signals, such as the presence of food [49]. In our study, a reduction in *ccnd1* expression was observed in the R3d group compared to the F4d group, while high levels of *ccnd1* were identified in the F15d and R6h groups compared to the C and R15d groups, respectively. The increased *ccnd1* expression in the F15d group may represent an adaptive measure that saves energy and substrates in the skeletal muscle of fish under fasting conditions. In senile human cells, this was reported to occur due to high levels of cyclin D1, which prevent DNA synthesis and delay cell cycle progression during the G1 phase [50,51]. This blockage of the cell cycle through *ccnd1* can compromise the proliferative processes required for the fish’s muscle growth. However, the progressive reduction in these cyclin levels resulting from refeeding, in both the short (R3d) and long (R15d) periods, may be associated with the nutritional restoration and continuity of the cell cycle, thus allowing muscle growth to resume. It should be noted that the lack of change in *ccnd1* expression in the F4d group compared to the C group may be associated with this less severe period of fasting, in which proliferative processes may not be affected. 

Regarding the final investigated target, a reduction in the *cdkn1a* (cyclin-dependent kinase inhibitor 1A, herein termed p21) levels was observed in the R3d group compared to the F4d group and in the R6h and R15d groups compared to the C and F15d groups, respectively. In addition to regulating the cell cycle, Cdkn1a acts on cell survival, with the mRNA of this gene expressed in both the liver and skeletal muscle of rats fasting for 24 and 48 h [34]. According to these authors, Cdkn1a regulates energy expenditure by controlling the lipid deposits and positively regulating PPARα (peroxisome proliferator-activated alpha receptor) [34]. PPARα is a crucial transcription factor that orchestrates numerous aspects of the adaptation to fasting, including fatty acid oxidation and ketogenesis [52]. In this context, our research group described high levels of *ppar* in the red muscle of the pacus during fasting, a muscle compartment that is less affected than white muscle [16]. Though we did not identify differences between the control and fasting groups in the case of *cdkn1a*, the literature data partially corroborate our results, demonstrating the important reduction in the *cdkn1a* levels during refeeding [38]. In this way, the maintenance of high levels of *cdkn1a* in the F4d and F15d groups, similar to the levels in the C group, may be associated with muscle adaptation to food restriction, and after food is made available to the muscle fibers, the transcriptional expression of *cdkn1a* decreases to very low levels. 

Microarray data provide unprecedented opportunities to explore new biological questions. Although our strategy has limitations associated with the low number of datasets available in the databases, the key genes selected in vitro were validated in pacus subjected to fasting and refeeding conditions, reinforcing the findings described here and the prospective power of the meta-analysis.

## 5. Conclusions

Our meta-analysis was an efficient strategy for the data mining and identification of non-classical genes associated with the maintenance of the skeletal muscle phenotype and growth in fish. Specifically, we demonstrated that ribosomal and cell cycle genes act in these processes in different species and under distinct experimental conditions. Furthermore, the investigation of the selected targets in the pacus demonstrated specific transcriptional modulations in short and extended fasting and refeeding periods, which can be associated with the maintenance of basal protein synthesis and tissue protection, together with energy saving. Together, our results help researchers to understand the adaptations of the skeletal muscle to the manipulation of food availability and are relevant for strategies that employ periods of fasting and refeeding in finfish farming and the natural environment.

## Figures and Tables

**Figure 1 genes-13-02378-f001:**
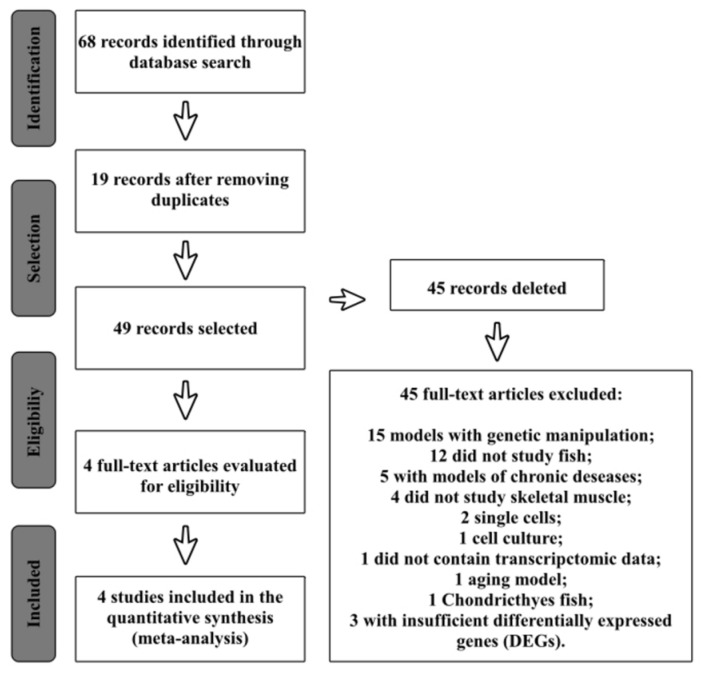
Literature search flowchart obtained from PRISMA in the meta-analysis.

**Figure 2 genes-13-02378-f002:**
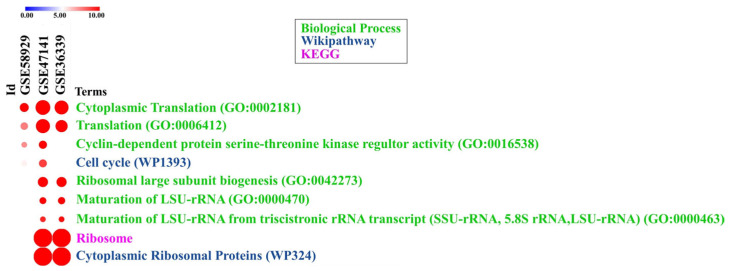
Biological processes and pathways associated with protein synthesis and proliferation were obtained from the enrichment of the pathways in each dataset, considering the Biological Process, WikiPathway, and KEGG libraries. The graph represents the selected processes that were present in at least two datasets. The GSE84288 dataset is absent from this graph due to the enrichment of the pathways that, unlike those of the other datasets, did not show predominant biological processes associated with protein synthesis and proliferation.

**Figure 3 genes-13-02378-f003:**
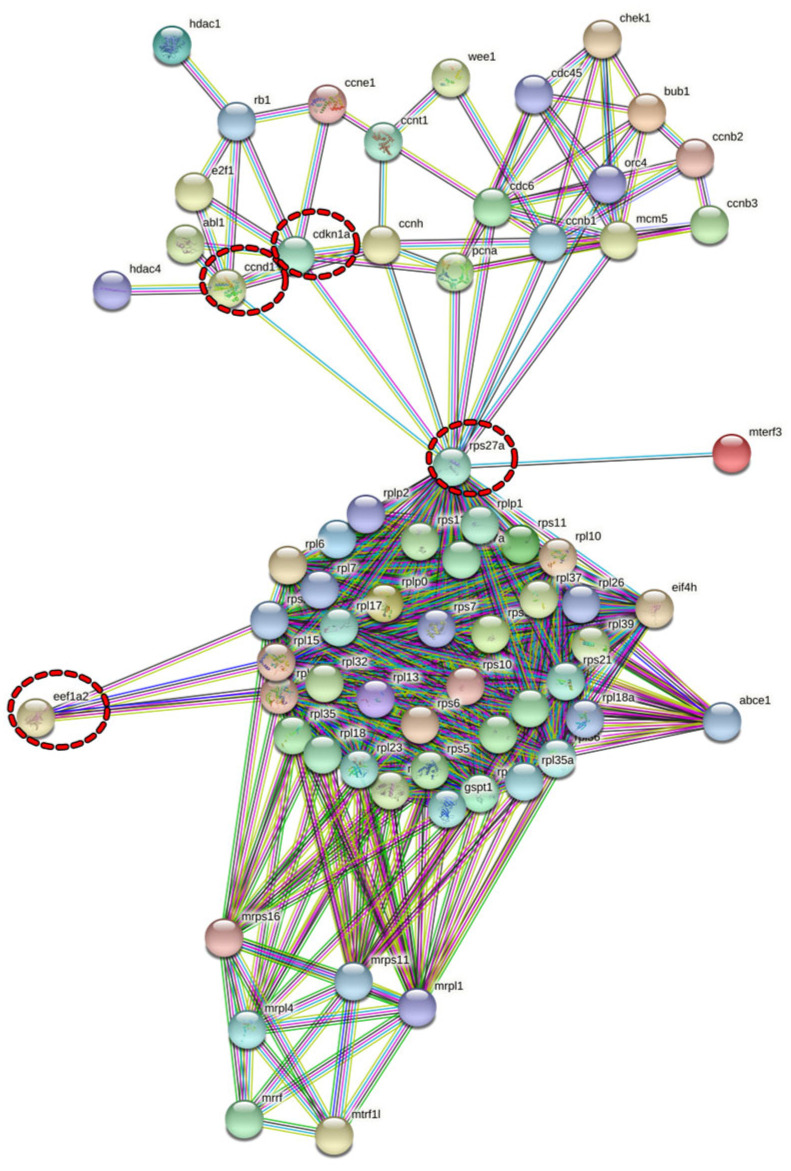
Molecular interaction network built on genes that enriched the cellular processes associated with proliferation and protein synthesis. The nodes represent the genes, and the edges represent the interactions between genes. The edge colors have different meanings: light blue: interactions from the curated databases; pink: interactions that were experimentally determined; green: neighborhood gene; red: gene fusions; dark blue: gene co-occurrence; lime: text mining; black: co-expression; and lilac: protein homology. In the upper region of the network are genes associated with the cell cycle, and in the lower portion are genes related to ribosomes. The dotted red circles highlight the genes selected for investigation: *rps27a*, *ccnd1*, *cdkn1a*, and *eef1a2*. The interaction score considered was 0.900.

**Figure 4 genes-13-02378-f004:**
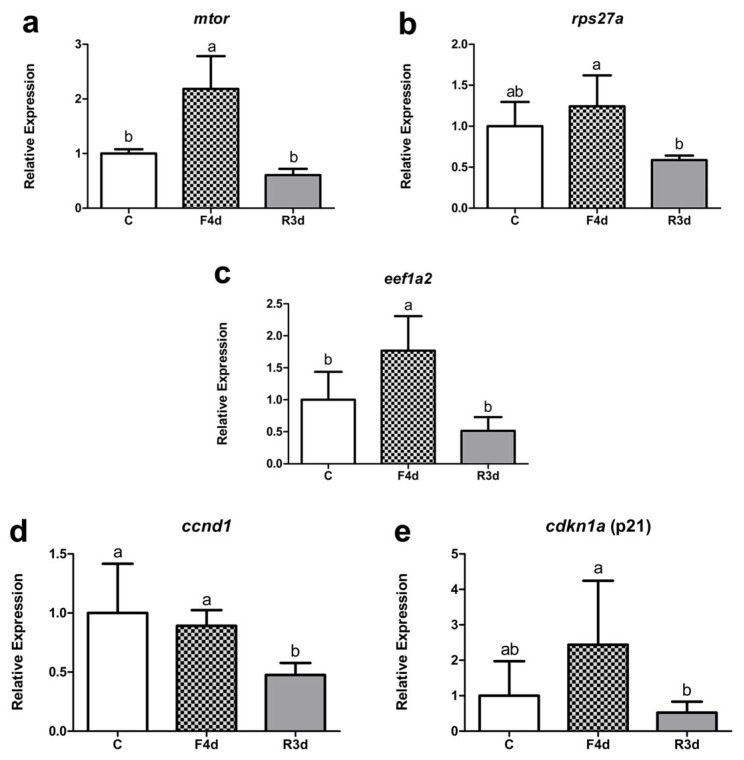
Relative mRNA expression of *mtor* (**a**), *rps27a* (**b**), *eef1a2* (**c**), *ccnd1* (**d**), and *cdkn1a* (**e**) in early juvenile pacu. The genes were evaluated in the control (C), four-day fasting (F4d), and three-day refeeding (R3d) groups. Data are expressed as the fold change of group C and presented as the mean ± SD. Different letters indicate significant differences between groups (*p* < 0.05 in a one-way ANOVA test, followed by Tukey’s multiple comparisons test for *mtor*, *rps27a*, *ccnd1*, and *cdkn1a*, or the Kruskal–Wallis and Dunn tests for *eef1a2*, n = 5/group).

**Figure 5 genes-13-02378-f005:**
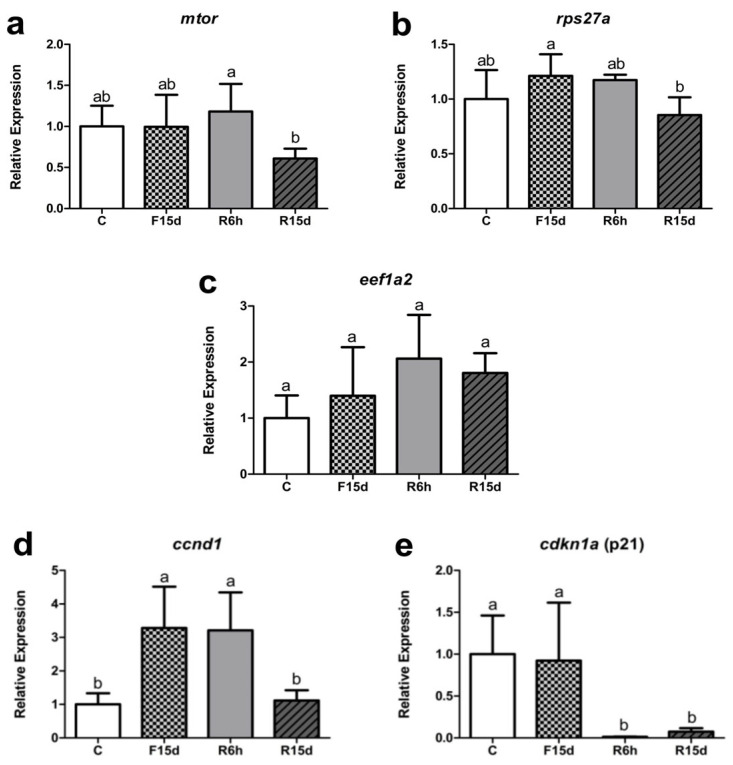
Relative mRNA expression of the *mtor* (**a**), *rps27a* (**b**), *eef1a2* (**c**), *ccnd1***(d)** and *cdkn1a* (**e**) in in late juvenile pacus. The genes were evaluated in the control (C), fifteen-day fasting (F15d), six-hours fasting (R6h), and fifteen-day refeeding (R15d) groups. Data are expressed as the fold change of group C and presented as the mean ± SD. Different letters indicate significant differences between groups (*p* < 0.05 in a one-way ANOVA test, followed by Tukey’s multiple comparisons test for *mtor*, *rps27a*, *ccnd1*, and *cdkn1a* or the Kruskal–Wallis and Dunn tests for *eef1a2*, n = 5/group).

**Table 1 genes-13-02378-t001:** Primers that were used for the amplification of *rps27a*, *eef1a2*, *ccnd1*, *cdkn1a*, *mtor*, and *ppia* mRNAs by RT-qPCR. F, forward; R, reverse. The genes are as follows: *rps27a* (ribosomal protein s27a), *eef1a2* (eukaryotic translation elongation factor 1 alpha 2), *ccnd1* (cyclin D1), *mtor* (mammalian target of rapamycin), *cdkn1a* (cyclin-dependent kinase inhibitor 1A), and *ppia* (peptidylprolyl isomerase A).

Symbol	Gene Name	Primer Sequence
*rps27a*	Ribosomal Protein S27a	F:CTACACCACCCCCAAGAAGA R:ATGAAAACACCAGCACCACA
*eef1a2*	Eukaryotic Translation Elongation Factor 1 Alpha 2	F:GAACAAATGCCACGGTTTCT R:AGAGCCCAACTACAGCCAGA
*ccnd1*	Cyclin D1	F:CACGATGCTAACCTGCTCAA R:TTTTGGGCACGATTTCTTTC
*cdkn1a*	Cyclin Dependent Kinase Inhibitor 1A	F: GTGGCGTTTCTCTGCTTAGG R:GATTCGGACTGAATGGCACT
*mtor*	Mechanistic Target of Rapamycin Kinase	F:TTGGGAGAGACGTACTGC R:CACAGGACTGGTGTAGGAA
*ppia*	Peptidylprolyl Isomerase A	F:ATTGTGGTTCGTGAAGTCGC R:CCGCTGGGCAGAGTGATTAT

**Table 2 genes-13-02378-t002:** Identification, characterization, and number of genes from each study included in the data analysis.

Serial Number	Species	Treatment	Upregulated Genes	Downregulated Genes
GSE58929	*Danio rerio*	Exercise	1516	215
GSE84288	*Salmo salar*	Consumption of glucosinolates	195	158
GSE36339	*Sparus aurata*	Consumption of lipopolysaccharides	43	38
GSE47141	*Oncorhynchus mykiss*	Exercise and a high-carbohydrate diet	152	206
**Total genes**			1906	617

## Data Availability

The datasets analyzed in the current study are available in the GEO dataset (https://www.ncbi.nlm.nih.gov/gds, accessed on 1 June 2020) repository. Each dataset has an access link: GSE58928 (https://www.ncbi.nlm.nih.gov/geo/query/acc.cgi?acc=GSE58929, accessed on 1 January 2020), GSE84288 (https://www.ncbi.nlm.nih.gov/geo/query/acc.cgi?acc=GSE84288, accessed on 1 January 2020), GSE36339 (https://www.ncbi.nlm.nih.gov/geo/query/acc.cgi?acc=GSE36339, accessed on 1 January 2020), and GSE47141 (https://www.ncbi.nlm.nih.gov/geo/query/acc.cgi?acc=GSE47141, accessed on 1 January 2020). Appendix A provided by the authors and contained in the original published articles were used for the analysis of the last two datasets. These original articles can be accessed at PMID: 22825392 (GSE36339) and PMID: 23968867 (GSE47141). All the data generated by our study are available throughout the text and in the Appendix A.

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
