# Peer review of "Identification of Novel Genes Associated with Fish Skeletal Muscle Adaptation during Fasting and Refeeding Based on a Meta-Analysis"

_genes, 2022, doi:10.3390/genes13122378_

Round 1

Reviewer 1 Report

The authors applied meta-analysis to reveal the muscle growth regulators of fish and provides new information on muscle responses to fasting and refeeding in fish of economic importance to aquaculture. The data presented by the authors are original and significant. The study is correctly designed and the authors used appropriate sampling methods. In general, statistical analyses are performed with good technical standards. The authors conducted careful work.

Specific remarks

L172: Why did the author choose the time of fasting (four or fifteen days) and the time of refeeding (six hours, three days or fifteen days later)? Is there any basis?

L185: How many samples were taken from each experimental group?

L313: Figure 4 and 5 can be displayed in one diagram; as can Figure 6 and 7.

L444: Change “of Eef1a2 improves” to “of eef1a2 improves”.

L478: Since cdkn1a is used in the full text, it is recommended not to use p21 to ensure consistency.

Author Response

Dear, Ms. Cathryn Zhang

Assistant Editor

Genes Journal - MDPI

We would like to thank the opportunity to submit a revised version of the manuscript (genes-2052642) entitled “Identification of novel genes associated with fish skeletal muscle adaptation during fasting and refeeding based on meta-analysis.” We have changed the manuscript as suggested by the reviewers. The new information and necessary corrections are clearly indicated in the manuscript, as recommended. We appreciate the careful attention of the reviewers and believe that this revised manuscript is substantially improved. Thank you for all your suggestions.

            In submitting this work for publication, we attest that the manuscript will not be submitted for publication elsewhere until a final decision regarding its acceptability has been made by the editors of Genes. We declare that all authors have approved the manuscript for submission and that there are no financial or non-financial competing interests.

            In addition to this document containing the answers to the reviewers, we also forward the English correction certificate provided by MDI. In addition to this document containing the answers to the reviewers, we also forward the English correction certificate provided by MDI. We inform you that we changed the title of the manuscript after the English correction. 

Thank you for your consideration.

Sincerely,

Dra Maeli Dal Pai

Department of Structural and Functional Biology

São Paulo State University, Botucatu, Brazil

E-mail: maeli.dal-pai@unesp.br

Answer to Reviewer 1:

The authors applied meta-analysis to reveal the muscle growth regulators of fish and provides new information on muscle responses to fasting and refeeding in fish of economic importance to aquaculture. The data presented by the authors are original and significant. The study is correctly designed and the authors used appropriate sampling methods. In general, statistical analyses are performed with good technical standards. The authors conducted careful work.

We thank you all for your constructive comments that provided valuable insights to refine its contents and analysis. In this document, we addressed the issues raised as best as possible.

1) L172: Why did the author choose the time of fasting (four or fifteen days) and the time of refeeding (six hours, three days or fifteen days later)? Is there any basis?

Thank you for the questions. In our research group, we have previously used 4 and 15 days of fasting and 6 hours, 3 days, and 15 days of refeeding to investigate the effects of these protocols on the morphology and expression of genes classically associated with anabolism and catabolism in fish skeletal muscle (10.3390/ijms22062995; 10.1371/journal.pone.0255006) (Methods, page 4, lines 187-191). These studies allowed us to understand the biology of skeletal muscle in the presence and absence of food, common aspects in nature, and also applied in aquaculture, aiming at the compensatory growth of fish. Thus, given the significant changes previously observed by our group, we choose these same times of fasting and refeeding to investigate the behavior of non-classical genes obtained through meta-analysis.

2) L185: How many samples were taken from each experimental group?

Thank you for pointing that out. We used 5 fish per experimental group. We regret that the word “samples” may have caused confusion. We have improved the text to clarify this point (Methods, page 4, Line 193).

3) L313: Figure 4 and 5 can be displayed in one diagram; as can Figure 6 and 7.

Thank you for your suggestion. The new figures 4 and 5 were added to the text accordingly to the reviewer's suggestion (Results, page 9, Figure 4; page 10, Figure 5).

4) L444: Change “of Eef1a2 improves” to “of eef1a2 improves”.

Thanks for pointing this out. We have replaced the text according to your suggestion (Discussion, page 11, line 463).

5) L478: Since cdkn1a is used in the full text, it is recommended not to use p21 to ensure consistency.

We have replaced p21 throughout the text (highlighted in red) to ensure consistency.

Reviewer 2 Report

The theme of this study is interesting, and the manuscript structure is reasonable. The analysis method is properly designed and provides reliable data in sufficient detail to support their conclusions. I suggest that the author has properly polished the language.

Author Response

Dear, Ms. Cathryn Zhang

Assistant Editor

Genes Journal - MDPI

We would like to thank the opportunity to submit a revised version of the manuscript (genes-2052642) entitled “Identification of novel genes associated with fish skeletal muscle adaptation during fasting and refeeding based on meta-analysis.” We have changed the manuscript as suggested by the reviewers. The new information and necessary corrections are clearly indicated in the manuscript, as recommended. We appreciate the careful attention of the reviewers and believe that this revised manuscript is substantially improved. Thank you for all your suggestions.

            In submitting this work for publication, we attest that the manuscript will not be submitted for publication elsewhere until a final decision regarding its acceptability has been made by the editors of Genes. We declare that all authors have approved the manuscript for submission and that there are no financial or non-financial competing interests.

            In addition to this document containing the answers to the reviewers, we also forward the English correction certificate provided by MDI. In addition to this document containing the answers to the reviewers, we also forward the English correction certificate provided by MDI. We inform you that we changed the title of the manuscript after the English correction. 

Thank you for your consideration.

Sincerely,

Dra Maeli Dal Pai

Department of Structural and Functional Biology

São Paulo State University, Botucatu, Brazil

E-mail: maeli.dal-pai@unesp.br

Answer to Reviewer 2:

1) The theme of this study is interesting, and the manuscript structure is reasonable. The analysis method is properly designed and provides reliable data in sufficient detail to support their conclusions. I suggest that the author has properly polished the language.

We are sorry that there were problems with English. A professional language editing service has carefully proofread the article to improve grammar and readability. We attach the review certificate and inform you that we changed the title of the manuscript after the English correction.

Reviewer 3 Report

1. Line-19, "(four or fifteen days) and refeeding (six hours, three or fifteen days)". The use of the word "or" makes it unclear and confusing.

2. Line-40-41, not according to the guidelines.

3. "1.2 Meta-analysis and data reanalysis" makes this manuscript confusing, is this the original manuscript research article or a meta analysis/literature study? I think it needs to be overhauled.

And if it's a meta analysis, then this is a systematic review that needs to be registered with PROSPERO-NIHR. I don't think it needs to be made into a separate section, but goes straight into one with the introduction (no subtitles).

4. This study is very confusing, there are in silico results, even though the objectives are not visible and the in silico method is not given in detail!

5. Line 267, the table title must be above the table, not below. Table 2 contains the typo "Specie".

6. Writing thousands must be accompanied by a comma, for example "1516" must be 1,516.

7. Line 305 "The interaction scores considered were 0.9 and 0.4. " What is the unit? A? or? and visualization of Figure 3 using what software?

8. Writing "mtor" terms must comply with the rules of MeSH, "mTOR". Check the entire manuscript and other terms. Font size must be considered!

Author Response

Dear, Ms. Cathryn Zhang

Assistant Editor

Genes Journal - MDPI

We would like to thank the opportunity to submit a revised version of the manuscript (genes-2052642) entitled “Identification of novel genes associated with fish skeletal muscle adaptation during fasting and refeeding based on meta-analysis.” We have changed the manuscript as suggested by the reviewers. The new information and necessary corrections are clearly indicated in the manuscript, as recommended. We appreciate the careful attention of the reviewers and believe that this revised manuscript is substantially improved. Thank you for all your suggestions.

            In submitting this work for publication, we attest that the manuscript will not be submitted for publication elsewhere until a final decision regarding its acceptability has been made by the editors of Genes. We declare that all authors have approved the manuscript for submission and that there are no financial or non-financial competing interests.

            In addition to this document containing the answers to the reviewers, we also forward the English correction certificate provided by MDI. In addition to this document containing the answers to the reviewers, we also forward the English correction certificate provided by MDI. We inform you that we changed the title of the manuscript after the English correction. 

Thank you for your consideration.

Sincerely,

Dra Maeli Dal Pai

Department of Structural and Functional Biology

São Paulo State University, Botucatu, Brazil

E-mail: maeli.dal-pai@unesp.br

Answer to Reviewer 3:

We thank you all for your constructive comments that provided valuable insights to refine its contents and analysis. In this document, we addressed the issues raised as best as possible.

1) Line-19, "(four or fifteen days) and refeeding (six hours, three or fifteen days)". The use of the word "or" makes it unclear and confusing.

Thanks for the suggestion. We have replaced the text accordingly (Abstract, page 1, line 16).

2) Line-40-41, not according to the guidelines.

Thanks for the suggestions. The text has been corrected according to the template.

3) 1.2 “Meta-analysis and data reanalysis" makes this manuscript confusing, is this the original manuscript research article or a meta analysis/literature study? I think it needs to be overhauled.

Thank you for this valuable feedback. We agree that our first version of this manuscript was confusing regarding these aspects. In fact, our study is a meta-analysis that confirmed the results using ours in vivo experimental model. Please note that, as clarified to reviewer 2, we have reworded the text to state that our research was based on a meta-analysis. We followed the Preferred Reporting Items for Systematic reviews and Meta-Analyses (PRISMA) statements (10.1136/bmj.n71), as can be seen in the PRISMA checklist included at the end of this document.

4) And if it's a meta analysis, then this is a systematic review that needs to be registered with PROSPERO-NIHR. I don't think it needs to be made into a separate section, but goes straight into one with the introduction (no subtitles).

Thanks to the reviewer's comment. We accessed PROSPERO-NIHR and, as described in this platform, our meta-analysis is not suitable to be registered according to the website. The main inclusion criterion for studies is that “PROSPERO includes details of any planned or ongoing systematic review with a health-related outcome”. Also, it is clearly described that “PROSPERO does not accept systematic reviews without results of clear relevance to human health”. However, seeking to meet the reviewer's request, in addition to PROSPERO, we consulted other platforms, such as IMPLASY and RESEARCH REGISTRY, which similarly prioritize studies aimed at humans. Also, a recently published meta-analysis (https://doi.org/10.3390/ani10071130) evaluated the levels of myoglobin in fish skeletal muscle. Similar to our study, this meta-analysis was not registered on online platforms.

5) This study is very confusing, there are in silico results, even though the objectives are not visible and the in silico method is not given in detail!

We regret there were problems with these topics. The paper has been carefully revised for clarity, the objective has been improved in the summary and introduction sections (page, lines). The in silico and in vivo studies complement each other. However, they were individualized and better detailed in the sections Methods and Results.

6) Line 267, the table title must be above the table, not below. Table 2 contains the typo "Specie".

Thank you for the observation. The title (Methods, page 5, lines 233-237) and the word line (Results, page 7, line 284) has been corrected in the manuscript.

7) Writing thousands must be accompanied by a comma, for example "1516" must be 1,516.

Thank you for the observation. We changed this point in the manuscript (Results, page 6, lines 260 and 262).

8) Line 305 "The interaction scores considered were 0.9 and 0.4. " What is the unit? A? or? and visualization of Figure 3 using what software?

Thank you for the questions. We agree with your doubt about the scores used and, for this reason, we modified the Figure 3 (Results, page 8, lines 309-310) using only the score 0.9. Figure 3 was generated using the STRING version 11 tool (https://string-db.org/). This tool provides interaction networks with protein-protein and during the assembly of the networks, there are some parameters that can be selected to make it even more reliable and, in this case, we chose the highest interaction score (0.9). This value is a score and therefore has no unit.

9) Writing "mtor" terms must comply with the rules of MeSH, "mTOR". Check the entire manuscript and other terms. Font size must be considered! (8)

Thanks for your observation. In Discussion, page 10, line 422, mTOR refers to the protein described in the cited article; on the other hand, in our study, we evaluated the mtor gene in fish. Although there is no specific regulation for the species of pacu (Piaractus mesopotamicus), in fish, genes are often written in lowercase and italics, as described in ZFIN Zebrafish Nomenclature Conventions. https://zfin.atlassian.net/wiki/spaces/general/pages/1818394635/ZFIN+Zebrafish+Nomenclature+Conventions

PRISMA CHECKLIST

Section/topic

#

Checklist item

Reported on page #

TITLE

Identification of novel genes associated with fish skeletal muscle adaptation during fasting and refeeding based on meta-analysis.

1

Meta-analysis

1

ABSTRACT

Structured summary

2

Regulation of the fish phenotype and muscle growth is influenced by fasting and refeeding periods, which occur in nature and are commonly applied to fish farming. However, the regulators associated with muscle responses to these manipulations of food availability are not fully characterized. We select novel genes associated with the growth and maintenance of the muscle phenotype, investigating them during periods of fasting and refeeding. Data were obtained from GEO Datasets using meta-analysis. Of the 68 total records found, 19 duplicates were removed, leaving 49. Of these, 45 were excluded based on the exclusion criteria: (1) studies with genetic manipulation, (2) studies that did not use fish as a model, (3) studies with models of chronic disease, (4) studies that did not assess skeletal muscle, (5) single -cell data, (6) cell culture data, (7) studies that did not contain transcriptomic data, (8) aging model, (9) studies with fish of the Chondrichthyes class, (10) studies with insufficient differentially expressed genes (DEGs), less than 10 genes. After, 4 full text articles were selected and their raw data and DEGs were downloaded and analyzed. For the analysis of data in the one-color system, the GEO2R tool (http://www.ncbi.nlm.nih.gov/geo/geo2r/) was used, which applies the same script (R 3.2. Biobase 2.30 .0, GEOquery 2.40.0, limma 3.26.8) for all datasets. To select differentially expressed genes (DEGs) in this system, adjusted p-values (padj) of < 0.05 and |fold change (FC)| > 1.5 were considered. For the analysis of the data in the two-color system, the DEGs were obtained through a list provided by authors who applied the difference of the log2 ER (expression ratio: Cy5/Cy3 or Cy3/Cy5) of zero between the chips to obtain these DEGs and from this list, we considered p < 0.05 for our DEGs. Between these DEGs obtained, some were related to translational and proliferative machinery and were investigated in pacus (Piaractus mespotamicus) subjected to fasting (four or fifteen days) and refeeding (six hours, three or fifteen days). Our results showed that different periods of fasting and refeeding modulate the expression of the genes mtor, rps27a, eef1a2, and cdkn1a. These alterations can indicate a possible muscle phenotype protection, in addition to adaptive responses that prioritize energy and substrate savings over cell division, a process regulated by ccnd1. Our study reveals the potential of meta-analysis in prospecting muscle growth regulators and provides new information on muscle responses to fasting and refeeding in fish of economic importance to aquaculture. Limitations may be associated with the microarray technique, which is not as embracing as RNAseq, for example. However, the analysis performed by this study provided robust data found in fish skeletal muscle, from different species and experimental conditions.

1 - 12

INTRODUCTION

Rationale

3

Meta-analysis is an excellent way to compile and explore data obtained by other researchers about a topic studied. As one of the main advantages for this study, the meta-analysis made it possible to obtain transcriptomic data relevant to the muscle biology of fish that have not yet been explored under the proposed conditions.

2, 10, 12

Objectives

4

To identify novel genes associated with fish skeletal muscle adaptation during fasting and refeeding based on meta-analysis.

1,2

METHODS

Protocol and registration

5

Not applicable

Eligibility criteria

6

The inclusion criterias were (1) studies with raw microarray data of mRNAs from the white and red muscle of fish and (2) only samples from fish of the Osteichthyes class. The exclusion criteria were (1) studies with genetic manipulation, (2) studies that did not use fish as a model, (3) studies with models of chronic disease, (4) studies that did not assess skeletal muscle, (5) single -cell data, (6) cell culture data, (7) studies that did not contain transcriptomic data, (8) aging model, (9) studies with fish of the Chondrichthyes class, (10) studies with insufficient differentially expressed genes (DEGs), less than 10 genes.

2,3

Information sources

7

This meta-analysis was performed with studies between 2012 and 2018, in which the data were accessed in 2020. The datasets analyzed during the current study are available in the GEO DataSets (https://www.ncbi.nlm.nih.gov/gds) repository. Each dataset has an access link: GSE58928(https://www.ncbi.nlm.nih.gov/geo/query/acc.cgi?acc=GSE58929), GSE84288(https://www.ncbi.nlm.nih. gov/geo/query/acc.cgi?acc=GSE84288),GSE36339(https://www.ncbi.nlm.nih.gov/geo/query/acc.cgi?acc=GSE36339) and GSE47141(https:// www.ncbi.nlm.nih.gov/geo/query/acc.cgi?acc=GSE47141). Supplementary tables provided by the authors and contained in the original published articles were used for the analysis of the last two data sets. These original articles can be accessed at PMID: 22825392 (GSE36339) and PMID: 23968867 (GSE47141).

14

Search

8

We used Pubmed to search the key-words “fish AND skeletal muscle”, “fish skeletal muscle AND transcriptome”, “fish muscle AND transcriptomic”, “fish skeletal muscle AND transcriptomics” e “fish skeletal muscle AND mRNA profiling”, all studies were selected considering all pages available in GEO Datasets for each key word. These studies were organized in sheets containing features of each study, including inclusion and exclusion criteria, date and author’s data collection method. 

2

Study selection

9

The selection of studies was performed manually by one researcher and reviewed by other authors of the work, independently. The data searched were organized in sheets containing essential characteristics that help to decide whether the study would be included or not.

not specified in the body of the manuscript

Data collection process

10

After obtaining the records, an initial assessment was carried out in the GEO Datasets to obtain information on how the studies were conducted by the researchers. In addition, the existence of the publication of the scientific article was verified based on these data, which were downloaded and analyzed using new statistical parameters. After, the data were analyzed through the GEO2R tool (http://www.ncbi.nlm.nih.gov/geo/geo2r/), which applies the same script (R 3.2. Biobase 2.30 .0, GEOquery 2.40.0, limma 3.26.8) for all datasets. To select differentially expressed genes (DEGs) in this system, adjusted p-values (padj) of < 0.05 and |fold change (FC)| > 1.5 were considered. For the reanalysis of the data in the two-color system, the DEGs were obtained through a list provided by authors who applied the difference of the log2 ER (expression ratio: Cy5/Cy3 or Cy3/Cy5) of zero between the chips to obtain these DEGs and from this list, we considered p < 0.05 for our DEGs. As a way to confirm the results obtained in silico, genes were investigated by RT-qPCR.

3

Data items

11

The array platform is an important variable, as it influences data processing, requiring different tools for analysis. Furthermore, the red and white skeletal muscles were considered together as a way to avoid few genes in the final analysis. Finally, all fish in the studies considered were at the same stage of development, being adults.

not specified in the body of the manuscript

Risk of bias in individual studies

12

The microarray is a customizable system and is made according to the research objective. Therefore, the transcriptomic profile is more limited on microarray data compared to RNAseq analysis. In this way, there is a risk of bias of the tool used in relation to the genes chosen for analysis against those expressed in the cells. To avoid bias of the individual studies, new statistical cut-off values were used for all analyzed data, the same values for all data sets. Furthermore, during the analysis, variables such as “time” were smoothed through data manipulation, for example, genes in common were selected between treatments with different times, leaving only the effect of the treatment itself.

not specified in the body of the manuscript

Summary measures

13

The measurements used were the value of p and Fold change.

3

Synthesis of results

14

The data of each study was extracted and analyzed using specific methods according to type data. Was used Geo2R (http://www.ncbi.nlm.nih.gov/geo/geo2r/) for One-Color microarray data and for the analysis of the data in the Two-colors system, the differentially expressed genes (DEGs)  were obtained through a list provided by authors who applied the difference in the log2 ER (expression ratio: Cy5/Cy3 or Cy3/Cy5). These results could be compared and compiled at the end, considering the same unit for all (p value and Fold Change).

3

Page 1 of 2

Section/topic

#

Checklist item

Reported on page #

Risk of bias across studies

15

No specific method was used to assess the bias of the results of each study, however, the data were analyzed using the same statistical parameters for all selected studies in an attempt to reduce the bias of the results of the meta-analysis itself. The fact that the microarray is a customizable system according to the objectives of each researcher may represent a risk of bias, but does not limit the robustness and veracity of the data generated by this platform, which were investigated through RT-qPCR by the current study.

not specified in the body of the manuscript

Additional analyses

16

RESULTS

Study selection

17

68 total records were found, 19 duplicates were removed, leaving 49. Of these, 45 were excluded based on the exclusion criteria. 15 studies excluded because they have genetic manipulation; 12 did not study fish; 5 with model of chronic disease; 4 did not study skeletal muscle; 2 single cell; 1 cell culture; 1 did not contain transcriptomic data; 1 aging model; 1 Chondrichthyes fish; 3 with insufficient differentially expressed genes (DEGs). In the end, 4 were included in the analysis.

6

Study characteristics

18

4 studies were included with, among them one that evaluated Danio rerio submitted to exercise, one color array (GSE58929), another that evaluated Salmo salar and consumption of glucosinolates, one color array (GSE84288); Sparus aurata and consumption of lipopolysaccharides, two colors array (GSE3633) and the last that evaluated Oncorhynchus mykiss during exercise and high-carbohydrate diet, two colors array (GSE47141). 

6

Risk of bias within studies

19

The microarray is a customizable system and is made according to the research objective. Therefore, the transcriptomic profile is more limited on microarray data compared to RNAseq analysis. In this way, there is a risk of bias of the tool used in relation to the genes chosen for analysis against those expressed in the cells. However, this is an inherent factor of meta-analyses when analyzing microarray data from other studies. On the other hand, the data were analyzed using the same statistical parameters for all selected studies in an attempt to reduce the bias of the results of the meta-analysis itself.

not specified in the body of the manuscript

Results of individual studies

20

GSE36339: In this study, it have evaluated the in vivo effects of lipopolysaccharide (LPS) on the white and red skeletal muscle transcriptome of the gilthead seabream (Sparus aurata) by microarray analysis at 24 and 72 hours after injection. In white muscle, the transcriptomic response was characterized by an up-regulation of genes involved in carbohydrate catabolism and protein synthesis at 24 hours and a complete reversal of this pattern at 72 hours. In red muscle, an up-regulation of genes involved in carbohydrate catabolism and protein synthesis was observed only at 72 hours after LPS administration. Interestingly, both white and red muscles showed similar consistent down-regulation of immune genes at 72 hours post-injection. However, genes involved in muscle contraction showed a general up-regulation in response to LPS in both types of muscle; GSE47141: It have evaluated the effects of moderate-intensity sustained swimming on the transcriptomic response of red and white muscle in rainbow trout fed a carbohydrate-rich diet using microarray and qPCR. Analysis of the red and white muscle transcriptome revealed significant changes in the expression of a large number of genes (395 and 597, respectively), with a total of 218 differentially expressed genes (DEGs) common for both muscles. A large number of the genes involved in glucose use and energy generation, contraction, development, synthesis and catabolism of proteins were up-regulated in red and white muscle. Additionally, DEGs in both muscles were involved in processes of defense response and apoptosis; GSE58929: Zebrafish were trained at low swimming speed (0.1 m/s; non-exercised) or at their optimal swimming speed (0.4 m/s; exercised). A significant increase in fibre cross-sectional area (1,290 ± 88 vs. 1,665 ± 106 μm2) and vascularization (298 ± 23 vs. 458 ± 38 capillaries/mm2) was found in exercised over non-exercised fish. Gene expression profiling evidenced the transcriptional activation of a series of complex networks of extracellular and intracellular signaling molecules and pathways involved in the regulation of muscle mass, myogenesis and angiogenesis, many (e.g. BMP, TGF, FGF, Notch, Wnt, MEF2, Shh, EphrinB2) not previously associated with exercise-induced contractile activity, and that recapitulate in part the transcriptional events occurring during skeletal muscle regeneration; GSE84288: Microarray analyses performed in the liver, muscle and distal kidney of salmon under high dose of GLs suggested massive tissue remodeling and reduction of cellular proliferation in skeletal muscle and liver. In the distal kidney, gene expression profiles induced under the high dose of GLs pointed to activation of anti-fibrotic responses. At the same time, prevalent activation of genes from the Phase-2 detoxification pathways could be considered part of beneficial effects. Multiple gene expression evidence suggested GLs-mediated iron/heme withdrawal response, including increase in heme degradation in muscle (up-regulation of heme oxygenase-1), decrease of its synthesis in liver (down-regulation of porphobilinogen deaminase) and increase in iron sequestration from blood (hepatic induction of hepcidin-1 and renal induction of intracellular storage protein ferritin).

not specified in the body of the manuscript

Synthesis of results

21

GSE58929: Danio rerio submitted to Exercise; 1516 (genes up regulated) and 215 (genes down regulated); GSE84288: Salmo salar submitted to Consumption of glucosinolates; 195 (genes up regulated) and 158 (genes down regulated); GSE36339: Sparus aurata submitted to Consumption of lipopolysaccharides; 43 (genes up regulated) and 38 (genes down regulated);GSE47141: Oncorhynchus mykiss submitted to Exercise and high-carbohydrate diet; 152 (genes up regulated) and 206 (genes down regulated).

6,7

Risk of bias across studies

22

The microarray is a customizable system and is made according to the research objective. Therefore, the transcriptomic profile is more limited on microarray data compared to RNAseq analysis. In this way, there is a risk of bias of the tool used in relation to the genes chosen for analysis against those expressed in the cells. However, this is an inherent factor of meta-analyses when analyzing microarray data from other studies. On the other hand, the data were analyzed using the same statistical parameters for all selected studies in an attempt to reduce the bias of the results of the meta-analysis itself.

not specified in the body of the manuscript

Additional analysis

23

DISCUSSION

Summary of evidence

24

The results obtained through the meta-analysis reinforce the association between the cell cycle and ribosomes, but in a context of muscle growth in fish. As a result, an interaction network was obtained that demonstrated this association between the two cellular components, in which from it and based on the literature, genes possibly important for muscle growth in fish were selected. These key genes (rps27a, mtor, eef1a2, ccnd1 and cdkn1a) were investigated in juvenile pacus submitted to different periods of fasting (F4 and F15) and refeeding (R6h and R3d), considering the importance of manipulating nutrient availability in the aquaculture scenario. Evaluation of the shorter periods of fasting and refeeding showed an increase in the expression of mtor and eef1a2 in the skeletal muscle of the early juveniles in the F4d group compared to the C and R3d groups, while the expression of rps27a was reduced in the R3d group compared to the F4d group. Considering the genes associated with the cell cycle, reduced expression of ccnd1 was observed in the R3d group; both compared to the C and F4d groups. The expression of cdkn1a (p21), meanwhile, was downregulated in the R3d group compared to F4d. Considering the genes related to the translational machinery, at the end of the 15-day fasting period (F15d), the mtor expression remained unchanged in the skeletal muscle of the late juvenile pacus compared to the C group. In contrast, the mtor expression was reduced in the prolonged refeeding group (R15d) compared to the R6h group. Regarding the rps27a ribosomal gene, there was a decrease in its expression in the R15d group compared to the F15d group. On the other hand, the expression of the eukaryotic elongation factor eef1a2 showed no change in expression between the different groups. Regarding cell cycle and replication regulators, the expression of ccnd1 transcripts showed an increase in the F15d group, as well as in the R6h group, compared to the C and R15d groups. Meanwhile, the expression of cdkn1a (p21) was downregulated both in the R6h and R15d groups compared to the C and F15d groups.

10-12

Limitations

25

Limitations may be associated with the number of fish skeletal muscle datasets available in the database. However, the analysis carried out by this study provided robust data found in the skeletal muscle of fish, from different species and different experimental conditions, which were investigated through RT-qPCR in pacus submitted to fasting and refeeding.

12

Conclusions

26

Our meta-analysis was an efficient strategy for data mining and identification of non-classical genes associated with the maintenance of the skeletal muscle phenotype and growth in fish. Specifically, we demonstrate that ribosomal and cell cycle genes act in these processes in different species and under distinct experimental conditions. Furthermore, the investigation of selected targets in the pacus demonstrated specific transcriptional modulations in short and extended fasting and refeeding periods, which can be associated with the maintenance of basal protein synthesis and tissue protection, along with energy savings. Together, our results help to understand the adaptations of skeletal muscle during the manipulation of food availability, which are relevant for strategies that employ periods of fasting and refeeding in finfish farming and natural environment.

12

FUNDING

Funding

27

This study was financed by the National Council for Scientific and Technological Development (National Council for Scientific and Technological Development; CNPq), grant numbers #306678/2021-7 and by the Coordination for the Improvement of Higher Education Personnel – Brazil (CAPES) – Finance Code 001, grant number 88887.502814/2020-00.

14

From:  Moher D, Liberati A, Tetzlaff J, Altman DG, The PRISMA Group (2009). Preferred Reporting Items for Systematic Reviews and Meta-Analyses: The PRISMA Statement. PLoS Med 6(7): e1000097. doi:10.1371/journal.pmed1000097

For more information, visit: www.prisma-statement.org.

Round 2

Reviewer 3 Report

Nice work in this revision!